# Investigation of the Low-Populated Excited States of the HIV-1 Nucleocapsid Domain

**DOI:** 10.3390/v14030632

**Published:** 2022-03-18

**Authors:** Assia Mouhand, Loussiné Zargarian, Anissa Belfetmi, Marjorie Catala, Marco Pasi, Ewen Lescop, Carine Tisné, Olivier Mauffret

**Affiliations:** 1Expression Génétique Microbienne, UMR 8261, CNRS, Institut de Biologie Physico-Chimique (IBPC), Université de Paris, 75005 Paris, France; assia.mouhand@gmail.com (A.M.); marjorie.catala@ibpc.fr (M.C.); 2Laboratoire de Biologie et de Pharmacologie Appliquée (LBPA), UMR 8113 CNRS, Institut D’Alembert, École Normale Supérieure Paris-Saclay, Université Paris-Saclay, 4, Avenue des Sciences, 91190 Gif sur Yvette, France; loussine.zargarian@ens-paris-saclay.fr (L.Z.); anissa_belfetmi@hms.harvard.edu (A.B.); marco.pasi@ens-paris-saclay.fr (M.P.); 3Institut de Chimie des Substances Naturelles, UPR2301 CNRS, Université Paris-Saclay, 1 av. de la Terrasse, 91198 Gif-sur-Yvette, France; ewen.lescop@cnrs.fr

**Keywords:** HIV-1, nucleocapsid, NCp7, NCp9, NCp15, NMR, CPMG, CEST, dynamic, low-populated state, dark-state

## Abstract

The nucleocapsid domain (NCd), located at the C-terminus of the HIV-1 Gag protein, is involved in numerous stages of the replication cycle, such as the packaging of the viral genome and reverse transcription. It exists under different forms through the viral life cycle, depending on the processing of Gag by the HIV-1 protease. NCd is constituted of two adjacent zinc knuckles (ZK1 and ZK2), separated by a flexible linker and flanked by disordered regions. Here, conformational equilibria between a major and two minor states were highlighted exclusively in ZK2, by using CPMG and CEST NMR experiments. These minor states appear to be temperature dependent, and their populations are highest at physiological temperature. These minor states are present both in NCp7, the mature form of NCd, and in NCp9 and NCp15, the precursor forms of NCd, with increased populations. The role of these minor states in the targeting of NCd by drugs and its binding properties is discussed.

## 1. Introduction

HIV-1 nucleocapsid proteins are involved in many functions during the replication cycle [1,2,3]. The nucleocapsid domain (NCd) is a small basic protein constituted of two zinc knuckles separated by a semi-flexible short linker and flanked by N- and C-terminal disordered parts [4]. This domain is encompassed in the Pr55^Gag^ precursor and exists in different forms of maturation during the HIV-1 replication cycle (Figure 1A), namely NCp15, NCp9, and ultimately NCp7, the mature form of the NCd. NCd, plays a role in the protection of the viral RNA in the virion and behaves as an essential nucleic acid chaperon protein in reverse transcription in the early steps of the infection [2,3,5].

NCp7 is composed of two zinc knuckles (Figure 1A) that exhibit the same coordination motifs (CCHC) and number of residues. However, the ZKs display different roles and involvements in most of the functions of NCd [2,6,7]. It has been shown that mutations in the N-terminal zinc knuckle (ZK1) are more deleterious for the nucleic acid chaperon activity of NCp7 than mutations in the C-terminal zinc knuckle (ZK2) [8,9]. Additionally, recent data showed that mutations in ZK2 are more detrimental for the recruitment of RNA inside Gag–RNA complexes than mutations in ZK1 [7,10,11]. Additional proof of the non-equivalence of the two ZKs of the NCd relies on their different sensitivities towards NCd inhibitors based on zinc ejectors [12,13,14]. These families of inhibitors, notably the mercaptobenzamide thioesters, perturb the zinc coordination and promote the formation of a covalent cysteine–thioester bond that triggers the zinc ejection [12,15,16]. The reactivity of the cysteine thiolate is strongly dependent on its environment [15,17,18] and is higher for ZK2 cysteines than for ZK1 ones, emphasizing that the environment is different in the two ZKs [12,16,17]. Interestingly, the same reactivity difference with mercaptobenzamide thioesters between ZKs is observed in the context of Gag [19], suggesting that the non-equivalence of the ZKs is conserved whatever the maturation step of the NCd.

Slight differences in sequence and structure can partially explain the different properties of ZKs. Each ZK contains a single aromatic residue, F16 in ZK1 and W37 in ZK2, exactly at the same position after the first cysteine involved in Zn^2+^ coordination (Figure 1A). Numerous structural and biophysical studies have highlighted the optimal stacking between W37 and unpaired guanosine, explaining the preference of binding NCp7 to nucleic acids (NA) containing unpaired guanosine [7,20,21,22]. It was thus proposed that ZK2 could play a major role in the specific recognition of NA sequences when compared to ZK1, due to the presence of W37 [7,23]. ZK1 presents a larger hydrophobic platform than ZK2 that offers a larger surface of contacts with NA. It also has a larger electrostatic interaction component with RNA than ZK2. This can explain the more important role of ZK1 in NA destabilization than ZK2 [21,24,25,26].

Besides, it was also shown that the intrinsic structural and dynamical properties of the NCd also play roles in the differences of function of the two ZKs. Intrinsic dynamics of the NCd entail a difference in the reorientation properties of the two zinc fingers, a property that is related to the ZK accessibilities and their NA binding capabilities [7,21,23,27,28,29]. Two types of NMR experiments, CPMG relaxation dispersion [30] and CEST (chemical exchange saturation transfer) [31,32], can be used to probe the exchange processes between a major NMR state (observable or “visible”) and sparsely-populated states that are characterized by different NMR chemical shifts relative to the major state. These experiments allow researchers to identify and characterize minor conformers with lifetimes in the 0.1–50 ms range, and, in particular, the CEST method is well suited to investigate slower processes in the 3–50 ms range [31]. Quantitative analysis of these experiments can be done via propagation of the Bloch-McConnell equations and gives the kinetic rate constants for the exchange processes, the relative populations, as well as the chemical shifts of the invisible minor states. Using these experiments, two minor states were uncovered for the NCd [33]. They arise from the default of coordination of the Zn^2+^ ion in ZK2 due to the hydrolysis of a coordination bond between the Zn^2+^ ion and cysteine or a histidine residue from the CCHC motif in ZK2. Such defects were not detected in ZK1; it was thus proposed that the existence of these minor states in ZK2 explain the selective targeting of ZK2 by zinc ejectors.

In this paper, we further investigated these low-populated states of the NCd, in its different states of maturation, to determine in which conditions they appear, using ^15^N-CPMG relaxation dispersion and ^15^N-CEST NMR experiments. Finally, we discuss their potential roles on the function of NCd and their importance for therapeutic application.

## 2. Materials and Methods

### 2.1. Expression and Purification of Recombinant HIV-1 NCp7, NCp9, and NCp15

NCp7, NCp9, and NCp15 (HIV-1 strain NL4-3) were expressed without any tag either at their N- or C-termini in Escherichia coli from plasmids built into a pET-3a vector (Novagen). NCp7 and NCp9 were overexpressed in E. coli BL21(DE3)pLysE strain, while NCp15 was overexpressed in E. coli BL21(DE3)star. NCp7 was overexpressed isotopically ^15^N-labeled as previously described [20,34]. The same protocols were used for NCp9. For NCp15, the protocols of purification were modified since NCp15 presented a lower level of expression and was more sensitive to proteolysis [27]. The three proteins were concentrated and dialyzed against the NMR buffer (25 mM deuterated sodium acetate pH 6.5, 50 mM NaCl, 1 mM ZnCl_2_, and 0.1 mM β-mercaptoethanol) using an amicon unit of 3 kDa (Millipore). Otherwise stated, the final protein concentrations were 1 mM for NCp7, 0.8 mM for NCp15, and 0.44 mM for NCp9.

### 2.2. NMR Experiments

^15^N-CPMG relaxation dispersion experiments were recorded on Bruker 700 MHz and 950 MHz spectrometers equipped with cryogenic probes, at 10, 25, and 35 °C, and in the same conditions using the pseudo-three-dimensional (interleaved) constant time CPMG (Carr-Purcell-Meiboom-Gill) optimized as previously described [30,35]. The constant time CPMG delay (T_CPMG_) was established to be 40 ms. Twenty experiments were acquired with ^15^N 180° pulse of duration 100 μs, and with repetition frequencies varying between 25 and 1000 Hz. A ^1^H spin-lock was applied at a field strength of approximately 13 kHz during the ^15^N-CPMG sequence to suppress ^1^H-^15^N scalar coupling evolution and exchange between in-phase and anti-phase magnetization components; the exact field strength slightly varied (~10%) for different values of CPMG frequency as previously described [30]. The NMR experiments were processed using NMRPipe [36], and SPARKY [37] was used to measure the intensity values of cross-peaks for the different relaxation delays. CurveFit (AG. Palmer Lab) and homemade Python (www.python.org, accessed on 14 February 2022), R (www.r-project.org, accessed on 14 February 2022) scripts were used to determine the R_1_ and R_2_ relaxation rates, as well as the associated uncertainties from the single-exponential decay.

The ^15^N-CPMG peak intensities were converted into relaxation rates using procedures previously described [30,38]. The effective transverse relaxation (R_2eff_) at the lowest frequency CPMG and the R_2eff_ at the highest frequency CPMG was calculated using the equation: $R_{2eff}=\frac{1}{T_{\mathrm{cpmg}}}\times ln(\frac{I\left(0\right)}{I\left(\nu_{\mathrm{cpmg}}\right)}$ where *I* (0) is the intensity in the absence of relaxation time and *I* (ν_cpmg_) is the intensity at CPMG frequency with $T_{\mathrm{CPMG}}$ relaxation time. The exchange contributions to the relaxation decay (R_ex_) reported in Figure 1 were obtained using the equation $R_{ex}=R_{2}\left(\frac{1}{\tau_{cp}}\to 0\right)-R_{2}\left(\frac{1}{\tau_{cp}}\to \infty \;\right)$, [39] where $\tau_{cp}$ is the delay between 180° pulses in the CPMG pulse train, and $\frac{1}{\tau_{cp}}\to \infty$ is approximated by $\frac{1}{\tau_{cp}}=1000$.

The ^15^N-CPMG relaxation dispersion experiments with NCp7 samples at different concentrations (400 and 100 μM) were recorded at 950 MHz and 35 °C using 11 points with different ν_CPMG_ values. Τhe errors were evaluated using the formula $\;\sigma_{R2}=\frac{1}{T_{relax}}\sqrt{{\left(\frac{\sigma_{I1}}{I_{1\left(\omega_{1}\right)}}\right)}^{2\;}+{\left(\frac{\sigma_{I0}}{I_{0}}\right)}^{2}}$, (https://www.nmr-relax.com/refs.html (accessed on 14 February 2022), relax Manual [40]) where uncertainties on peak intensities, $\sigma_{I1}$ and $\sigma_{I0}$, were estimated using two repeated points in the CPMG experiments; *I*_1__(ω1)_ and *I*_0_ are the peak intensities at a frequency ω_1_ in the CPMG series, and *I*_0_ is the experiment performed without the constant time [30]. Only for the experiments with dilute samples, were the quantity of D2O in samples reduced to 4%, to decrease deuterium isotopic effects that affect the exchange rates [41].

The CEST experiments were recorded on the ^15^N-labeled NCp7 protein at 700 and 950 MHz and at 25 and 35 °C. The data recorded at 700 MHz comprised a series of 2D spectra with the ^15^N offset ranging between 102.2 and 132.61 ppm and obtained with increments of 0.41 ppm (25 Hz) or of 0.205 ppm (12.5 Hz). The ^15^N B1 field applied during the relaxation time (T_ex_) was 25 and 12.5 Hz, respectively. The T_ex_ was set to 0.4 s. At 950 MHz, the data were recorded as 2D spectra with the ^15^N offset ranging between 103.3 and 133.41 ppm and obtained with increments of 0.5 ppm (40 Hz) and 0.25 ppm (20 Hz). In agreement with these frequency increments, the ^15^N B1 field forcefield applied during the relaxation time (T_ex_ = 0.4 s) was 40 and 20 Hz, respectively.

CPMG and CEST relaxation data were analyzed with two-state and three-state models using the program ChemEx (https://github.com/gbouvignies/chemex, accessed on 14 February 2022), which numerically integrates the Bloch-McConnell equations [42] and allows the extraction of k_ex_AB_, k_ex_AC_, and k_ex_BC_ (kinetic rates of exchange between states A and B, A and C, and B, and C respectively), p_A_, p_B_, and p_C_ (populations of the different conformers), and ∆*ω*_AB_ and ∆*ω*_AC_ (chemical shift differences between the major and minor conformers). In the case of a three-state equilibrium between a major state A and two minor states B and C (p_A_ >> p_B_ or p_C_), the three states could mutually interconvert (Appendix A). These values were extracted for each residue of ZK2 in NCp7, NCp9, and NCp15. The fits are global, multi-residue, and multi-field (700 and 950 MHz).

A large grid search was used for the starting values of the following parameters: p_A_, p_B_, p_C_, k_ex_AB_, k_ex_AC_, and k_ex_BC_, ∆*ω*_AB,_ ∆*ω*_AC_. The parameter values giving the lowest $\chi^{2}\left(\xi \right)=\overset{N}{\underset{i=1}{\sum }}\overset{2}{\underset{j}{\sum }}{\left(\frac{I_{i}^{exptl}\;-\;I_{i}^{calcd}\left(\xi \right)}{\sigma_{i}^{exptl}}\right)}^{2}$ were found, where *i* is related to the number of residues (N = 10), *j* is related to the two fields used for the CPMG experiments (700 and 950 MHz), $I_{i}^{exptl}$ and $I_{i}^{calcd}\left(\xi \right)$ are the experimental and calculated peak intensities, the calculations are performed using $\xi =\left\{x_{1\;},\;\ldots ,x_{n}\right\}$ that refer to the different fitted parameters (p_A_, p_B_, p_C_, k_ex_AB_, k_ex_AC_, k_ex_BC_, etc. …), and $\sigma_{i}^{exptl}$ is the error in the measurement of intensities. To find the values presented in the tables, the calculation was repeated 100 times with a randomized Gaussian variation (associated with the rms noise measured in the spectra) applied to the intensity values of CPMG data to obtain the mean and standard deviations of the exchange parameters.

Mean ^15^N relaxation rates were calculated for residues of ZK1 or for residues of both ZK1 and ZK2 (ZK1 + ZK2), and compared using a two-sided paired Wilcoxon test.

As the measurements were made at 35 °C, fast hydrogen exchange occurs particularly in disordered regions of the proteins, i.e., the N- and C-termini domains. As mentioned in several publications, this may lead to R_ex_ contributions that are associated with hydrogen exchange, but not conformational exchange [41,43]. These artefactual R_ex_ exchange processes have been found to be related to deuterium isotopic effects resulting from solvent exchange occurring at timescales for which the CPMG experiments are sensitive [41]. In our case, the N- and C-terminal residues, which are largely disordered, exchange faster with solvent, particularly at 35 °C, leading to the detection of R_ex_ in ^15^N-CPMG dispersion relaxation experiments. We carefully examined these effects by using: (i) the fact that the artificially high R_ex_ are only observed at low CPMG frequencies (<100 Hz) [41]; (ii) sequence pulses classically used for T2 relaxation measurements with CPMG pulse trains during the relaxation delay period, and for which the use of certain CPMG frequencies (250 Hz) eliminate exchange contribution.

## 3. Results and Discussion

### 3.1. NCp7 and Its Precursors NCp9 and NCp15 Populate Two Minor States under Specific Temperature Conditions

In a first analysis, the ^15^N-CPMG relaxation dispersion profiles of NCp7, measured at 700 MHz in various conditions (two temperatures: 10 °C and 35 °C; three pH: 5.5, 6.0, and 6.5; two NaCl concentrations: 25 and 50 mM), were used to determine the R_ex_ values in order to investigate the NCp7 dynamics in the microsecond-to-millisecond timescale (Figure 1). This allowed us to rapidly determine the conditions where NCp7 undergoes conformational exchange.

The temperature is the major factor that promotes the appearance of conformational exchanges (Figure 1). Indeed, near the physiological temperature (35 °C), high values of R_ex_ were observed in ZK2, whereas all the residues of ZK1 exhibit rather flat dispersion profiles (Figure 1C,E). At 10 °C, no conformational exchange was observed in either ZK1 or ZK2 (Figure 1B,D). Small conformational exchanges began to be observed in ZK2 at 25 °C (data not shown), but it was really around physiological temperature that these exchanges became prominent. Large R_ex_ values also appear in flexible parts of NCp7, at the N- and C-termini, mainly at 35 °C and pH 5.5. These large R_ex_ values are quite probably related to the amide protons in fast exchange with water and are generated by deuterium isotopic effects as described previously [41]; this conclusion is reinforced by the analysis of other experiments as described in Materials and Methods. The ^15^N-CPMG NMR experiments on NCp9 and NCp15 were measured at pH 6.5 and 35 °C (Figure 2A,B); at these conditions of temperature and pH, most of the amide groups of the N-terminal part were not observable in the CPMG experiments.

As for NCp7, conformational exchange characterized by large values of R_ex_ (~20 s*^−^*^1^) was measured for residues in ZK2 of both NCp9 and NCp15 at 35 °C. Figure 2C shows the CPMG profiles for ZK2 residues that exhibit large R_ex_ values. The profiles for each residue are similar in NCp15 and NCp9, suggesting that the underlying mechanisms of conformational exchange are the same.

We further analyzed the data obtained for ZK2 residues of NCp7, NCp9, and NCp15 in the same conditions (i.e., 25 mM Na-acetate pH 6.5, 50 mM NaCl, 1 mM ZnCl_2_) at two magnetic field strengths (700 MHz and 950 MHz) using the ChemEx program, which allows a full and quantitative determination of the parameters associated with conformational exchange [31,32]. All the ^15^N-CPMG data (10 residues, located in ZK2: C36, W37, K38, C39, G40, E42, H44, M46, D48, C49) were fitted simultaneously with two different types of model, either with one or two minor conformers. When the model contains only one minor state, the fit of the data is poor for residues that show large R_ex_ values, such as C39 and W37. The addition of a third species with possible interconversion between the two minor states led to much more satisfying fits (Appendix A). The gain on fit quality is attested by extremely favorable F-statistics (Ftest = 34.7, pval = 3 × 10*^−^*^152^), as well as the visual inspection of the fits (Appendix A, for residue W37 and C39). Finding the best parameters required a large grid search for the (k_ex_AB_, k_ex_BC_, k_ex_AC_, p_B_, p_C_) parameters after a first optimization of the Δω values. The conformational landscape thus appears rather complex and extensive grid searches are needed to find the global minimum. The kinetic rate constants and the populations associated with the best Δω values are shown in Table 1. Fitting the ^15^N-CPMG profiles for ZK1 residues did not show the existence of minor forms, confirming that the low values of R_ex_ observed in ZK1 stem from the isotopic effects related to water exchange (see Materials and Methods). Adding data from residues of ZK1 in the fit did not change the values of populations and kinetic exchange rates for ZK2. In any case, the Δω values for the residues in ZK1 are extremely low, indicating no significant difference between the major and minor conformers. For NCp7, the Δω values are very close to that measured in the previous study from the Clore group [33], suggesting that the same minor states are observed.

The Δωs associated with the best fits for each of the three A, B, and C states are identical for the three proteins, showing that the same conformational exchanges with the same minor states are experienced whatever the step of maturation of the NCd (Figure 3). For the three proteins, ZK1 is thus conformationally stable and ZK2 interconverts on the millisecond timescale between the major state A, in which the zinc is coordinated by all the residues of the CCHC motif, and the two minor states B and C, in which one of the coordination bonds is hydrolyzed, as previously described for NCp7 [33]. State C is characterized by smaller chemical shift variations (Figure 3B) and is thus closer to the conformation of the major form than state B. This state C was previously proposed to be associated with the hydrolysis of the H44(N*ε*2)-Zn bond, whereas state B leads to more substantial and numerous chemical shift variations in ZK2 when compared to the major form (−8.0 ppm for W37, 5.5 ppm for C39, and 7.1 ppm for H44), and this would correspond to the hydrolysis of the C36(S*γ*)- or C39(S*γ*)-Zn bond. Given the large chemical shift variations between states B and A, we propose that state B corresponds to the hydrolysis of these two bonds, which could explain these large variations of Δ*ω*, as well as the small changes of backbone conformation previously observed for ZK2 residues [33]. States B and C are equally populated for NCp7 (2.7%), but state B is more populated for NCp9 and NCp15 (around 9%) and twice more populated than state C (Table 1). This suggests that p1, located just after the C-terminal of ZK2 in NCp9 and in NCp15, significantly modulates the exchange processes in ZK2, with a stabilizing effect of the de-coordinated minor states. In contrast, the additional presence of p6 in NCp15 did not further affect the conformational dynamics of ZK2. We previously showed that the p1-p6 region of NCp15 is largely disordered but interacts transiently with the ZKs [27]. These transient interactions notably involve the residues W37 and K38 of ZK2, as well as the hydrophobic residues of p1 (L57 and I60).

### 3.2. CEST Associated with CPMG Experiments on NCp7 and Analysis of the ^15^N Transverse Relaxation Rates

CEST experiments [32] have been recorded on NCp7 in similar experimental conditions to better determine the kinetic rate constants and chemical shift variations associated with the conformational exchange. As expected, the large chemical shift variations for C39, G40, Q45, and W37 were also observed in the CEST experiments with an excellent quantitative agreement, since the CEST dips appeared at the position expected from CPMG experiments (Figure 4). Three CEST dips, indicating a major form in exchange with two minor forms, can be clearly observed for Q45 (Figure 4). For all the other residues in exchange, only one single minor peak can be observed. In most cases, the second minor peak was not observed because it resonates very close to the major state A. CEST data can be used alone or in combination with CPMG data to complete or improve the characterization of exchange parameters. Previous works have shown that the determination of exchange parameters using CPMG methods could be difficult when the exchange rate decreases towards the slow regime (as it is the case here with k_ex_ ≈ 400 s^−1^), but adding CEST data can overcome this difficulty [31,32,44]. To correctly evaluate the contribution of CEST data relative to the CPMG data in the analysis with ChemEx, calculations were performed with only CEST or only CPMG or both together. The resulting fitted values of Δω and of the populations of the major and minor states are close to the results obtained with only the CMPG data. However, we do note that the k_ex_ values are slightly larger in the combined fits, particularly for k_ex_BC_ (Table 2).

All fitted parameters associated with the exchange processes that were extracted using chemEx (p_A_, p_B_, p_C_, k_ex_AB_, k_ex_AC_,and k_ex_BC_, Δ*ω*_AB,_ Δ*ω*_AC_) are similar to those previously measured for NCp7 [33], particularly the values for Δω, confirming that the same minor conformers, originating from a defect in the coordination of the zinc ion in ZK2, were observed.

The results of the calculations are shown in Table 2 and Appendix A. The calculations were performed with different sets of data, including only CEST data for residues featuring the largest Δ*ω*_AB_, either a set of three (CEST_H44,W37,C39_) or five (CEST_H44,W37,C39,G40,E42_) residues. CPMG data was included for three residues (CPMG_H44,W37,C39_) or for a larger set of 10 ZK2 residues (CPMG_10residues_). In Table 2, the populations and kinetic rate constants obtained with the different sets of data are similar, with a major state A populated at around 94–95% and two minor states quasi-equally populated at ~2% each, showing that the different sets of data converge to the same results. Concerning the kinetic rate constants, only one significant difference was observed for the rate constant k_ex_BC_, which is is higher for the CEST data. We think that the CEST data contain more information since the minor species were explicitly observed. Therefore, kinetic rate constants determined using CEST data that are notably related to the interconversion between the minor species, k_ex_BC_, are more accurate.

CEST experiments also permit researchers to evaluate the relaxation rates R2 of the individual states, in contrast to the calculations using only CPMG data that are made with the hypothesis that the intrinsic relaxation rates of the various states are similar [31,32]. The CEST-derived ^15^N intrinsic R2 values have been determined for the three states A,B, and C (Appendix A). In these calculations, upper and lower bounds (100 and 5 s*^−^*^1^) have been initially assigned for the R2 values. After calculations, the B state for residues W37, C39, and H44 was found to be associated with a R2 value of 100 s*^−^*^1^, much larger than the R2 values for states A and C (in the range 5–10 s*^−^*^1^). The state B was directly observed for these residues in the CEST experiments (Figure 4) and corresponds to the observable minor form that displays large broadening. As the broadening associated with the exchange between the three forms was appropriately accounted for in our calculations, the fact that we still observed high large R2 values for these residues is intriguing. An intrinsic large transverse rate restricted to some residues in a protein domain is evocative of additional exchange events occurring in this area that were not taken into account in the calculations [32,33]. In our case, this suggests that the residues W37, C39, and H44 could be involved in additional exchange processes with four, five, or a larger number of states exchanging at faster timescales than both k_AB_ and k_BC_, and leading to line broadening in state B. This could reflect, for instance, the rather unstable nature of the zinc-binding site in the minor state B. Another possibility is that state B could be conducive to NC oligomerization, leading to higher transverse rates that would depend on the size of the oligomers.

Recently, several studies have described and highlighted interactions and interconversions between monomeric and oligomeric species [45,46]. To test this latter possibility, we performed CPMG experiments at two different NCp7 concentrations (100 and 400 μM). For three residues that are involved in conformational exchange (W37, E42, H44), the CPMG profiles are affected by the variation of the protein concentration (Figure 5). While the CPMG profiles of these residues still exhibited conformational exchanges, these processes appeared more pronounced at the higher concentration. In contrast, other residues of ZF2 (C36, K38, C39, G40, D48, C49) did not show a significant variation of their CPMG profiles with concentration. Similarly, the ZF1 residues (such as H23 shown in Figure 5) did not show any exchange at either concentration.

We also measured the ^15^N transverse relaxation at the higher CPMG frequency (1000 Hz) because, in this condition, the broadening associated with chemical exchange is considered to be nearly eliminated. The R2 relaxation rates obtained at this high CPMG frequency correspond to the population-weighted average rates over the three states, as described in previous studies [46]. The R2 values (at 1000 Hz) obtained using this approach at the different concentrations (for 100 and 400 μM samples) are quite close over the protein sequence, and statistical tests confirmed that the differences are not statistically significant. This suggests that if oligomerization processes really occur, the degree of oligomerization is necessarily low (dimer, trimer …).

Altogether, our measurements of the effect of NCd concentration on relaxation rates show that three residues exhibit a concentration-dependent conformational exchange, while the other residues do not. This suggests that the zinc de-coordination occurs in ZK2 even at low concentrations, and that this process is not concentration-dependent. In contrast, the observation of concentration-dependent R_ex_ for the residues W37, E42, and H44 could be related to transient contacts between monomers of different nature or the formation of dimeric complexes. We can also hypothesize that these contacts, which concern residues sensitive to zinc coordination, could be involved in zinc exchange between species. Furthermore, it is still unclear whether the changes in Rex associated with the increase in concentration are caused by a conformational change or rather by an increase in water exchange rates.

## 4. Conclusions

In this paper, we used NMR relaxation experiments, CPMG and CEST, to show that the nucleocapsid domain of Gag populates minor conformers in all three of its maturation forms (NC15, NCp9, and NCp7). These minor conformers correspond to conformational changes in ZK2 only, and their populations are increased in the precursor forms of the nucleocapsid domain; this is up to about 10% in total. This increase is likely due to the p1 region that is found at the C-terminal of ZK2 in NCp9 and NCp15, which seems to stabilize these minor conformers. It was previously shown for NCp7 that these minor states correspond to defects in zinc coordination of ZK2 [33] and that p1 makes transient interactions with the ZKs (and particularly ZK2) of the nucleocapsid domain in NCp9 and NCp15 [27]. This dynamic behavior of p1 seems, therefore, to favor the zinc de-coordinated forms of the nucleocapsid domain. These results agree with the fact that the mercaptobenzamide thio-esters, which have been shown to target specifically the minor forms of NCp7 [33], are also able to target Gag with a similar mechanism [47]. Our work also shows that the observations of these forms are a temperature-dependent phenomenon, the minor forms being not visible at 10 °C but easily observed at 35 °C, near the physiological temperature. This phenomenon is rather intriguing and is reminiscent of a previous work describing the strong temperature dependence of NCp7 dynamics [48]. In addition, numerous biochemical studies have shown that optimal chaperon activity of the protein is achieved at temperatures close to physiological [49,50,51]. The present observation of the minor forms at a temperature close to physiological suggests that these forms could be involved in nucleic-acid chaperoning activity of the nucleocapsid domain.

The zinc de-coordination process is limited to ZK2, which is coherent with the fact that zinc binding affinity is higher for ZK1 than for ZK2 [52,53]. We could therefore envision that the slightly lower stability of zinc in ZK2 entails its partial de-coordination. Additionally, it has been shown that zinc plays an important role in RNA recognition by NC; in particular, contacts between ZK2 and RNA are more dependent on zinc concentration than ZK1–NA contacts [52]. Thus, we could hypothesize that these minor forms featuring partial zinc de-coordination have not only a consequent role in their recognition by NC inhibitors such as mercaptobenzamide thioesters, but could also play specific roles in the nucleic acid binding of NCd.

## Figures and Tables

**Figure 1 viruses-14-00632-f001:**
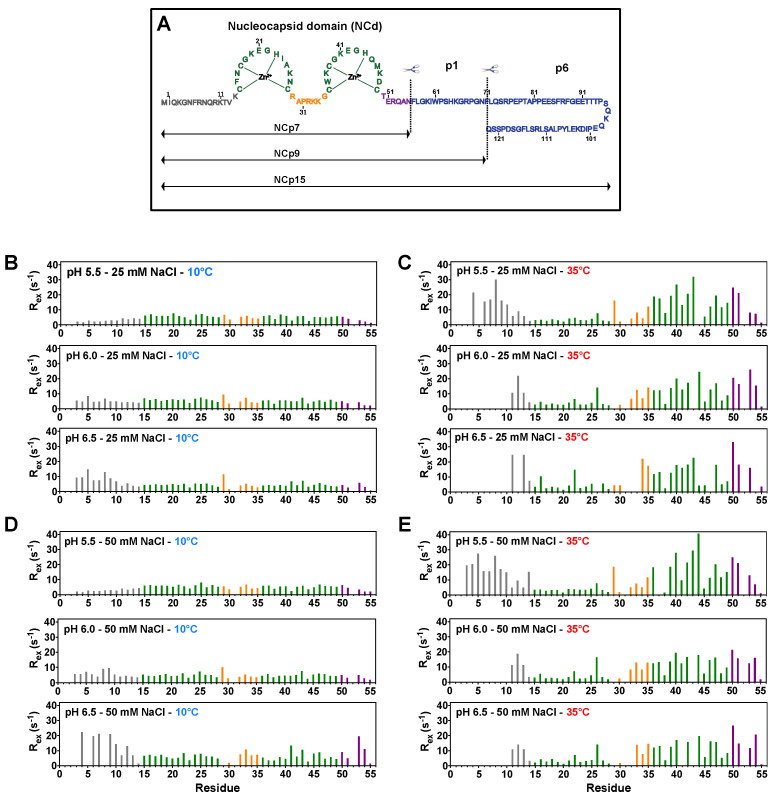
^15^N-CPMG relaxation dispersion experiments measured on NCp7 at 700 MHz and different conditions of temperature (10 °C and 35 °C), salt concentration (25 or 50 mM) and pH (5.5, 6.0, or 6.5). (**A**) Sequence of the HIV-1 C-terminal domain of Gag (NCp15). The first cleavage by the HIV-1 protease first liberates the NCp15 protein, then NCp9. and finally, the mature form of NCd called NCp7. The dashed line represents the two sites of protease cleavage present in NCp15. Residues are colored in grey for residues in the N-terminal part of NCd, green for those in the ZKs. orange for residues in the linker between the two ZKs of NCd, and purple for the C-terminal domain of NCd; (**B**) R_ex_ measured for NCp7 at 25 mM NaCl and 10 °C for three pH (5.5, 6.0, and 6.5); (**C**) R_ex_ measured for NCp7 at 25 mM NaCl and 35 °C for three pH (5.5, 6, and 6.5); (**D**) R_ex_ measured for NCp7 at 50 mM NaCl and 10 °C for three pH (5.5, 6, and 6.5); (**E**) R_ex_ measured for NCp7 at 50 mM NaCl and 35 °C for three pH (5.5, 6, and 6.5). In panels (**B**)–(**E**), the bars are colored using the same color code as the amino acid sequence in panel (**A**).

**Figure 2 viruses-14-00632-f002:**
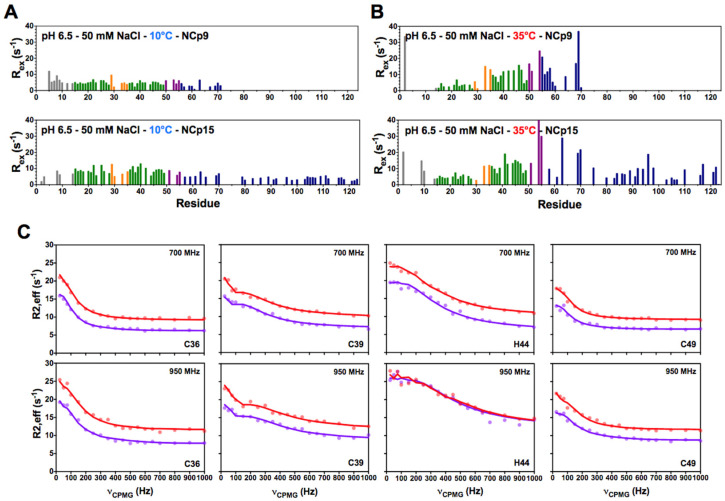
^15^N-CPMG relaxation dispersion experiments measured for NCp9 (upper panel), and NCp15 (lower panel) at 700 MHz, pH 6.5 and at (**A**) 10 °C or (**B**) 35 °C. (**C**) ^15^N-CPMG relaxation dispersion profiles of residues in NCp15 (in red) and NCp9 (in purple) that coordinate the zinc atom in ZK2 (C36, C39, H44, and C49). The experimental data are represented with dots and the fit curve of the data by lines. The fit was done with ChemEx for a three-state model, with one major and two minor states that interconvert.

**Figure 3 viruses-14-00632-f003:**
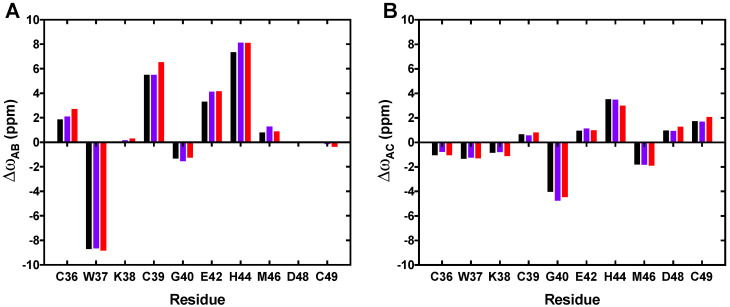
^15^N-chemical shift differences for ZK2 residues. (**A**) Δω_AB_ and (**B**) Δω_Ac_ between major and minor species obtained from the fit of ^15^N-CPMG relaxation dispersion experiment for NCp7 (black), NCp9 (purple), and NCp15 (red). The signs of Δω_AB_ and Δω_AC_ were obtained from the CEST data of NCp7 and the same signs were used to represent Δω_AB_ and Δω_AC_ of NCp9 and NCp15.

**Figure 4 viruses-14-00632-f004:**
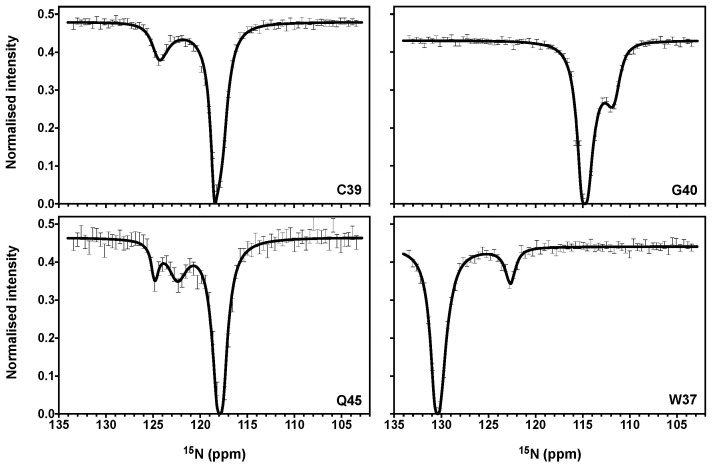
^15^N-CEST intensity profiles for C39, G40, Q45, and W37 residues obtained at 600 MHz and 25 °C with a B1 field of 25 Hz and T_ex_ = 0.4 s. The main dip is at the resonance frequency of the major state and minor dips were observed at the frequency of the minor states. For the figure, the ChemEx software was used to fit the data, using an individual fit for each residue.

**Figure 5 viruses-14-00632-f005:**
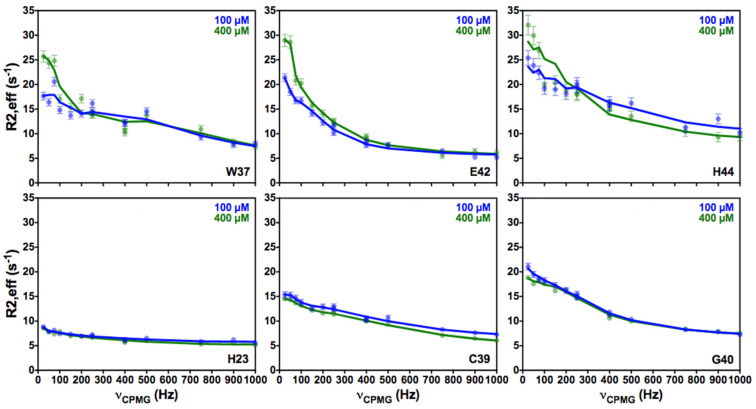
^15^N-CPMG relaxation dispersion profiles measured at 950 MHz and 35 °C at two different protein concentrations, 100 μM (blue) and 400 μM (green), for six residues of NCp7.

**Table 1 viruses-14-00632-t001:** Populations and kinetic rate constants extracted from global fits with ChemEx of the ^15^N-CPMG relaxation dispersion profiles for ZK2 residues (C36, W37, K38, C39, G40, E42, H44, M46, D48, C49) in NCp7, NCp9, and NCp15 at 35 °C and 700 MHz and 950 MHz. The mean values and standard deviations have been obtained using 100 Monte-Carlo calculations as described in Materials and Methods.

	NCp7	NCp9	NCp15
p_A_	0.945 ± 0.002	0.883 ± 0.009	0.870 ± 0.007
p_B_	0.027 ± 0.002	0.086 ± 0.008	0.087 ± 0.007
p_C_	0.027 ± 0.001	0.030 ± 0.001	0.041 ± 0.001
k_ex_AB (s_^−1^_)_	331 ± 31	66 ± 7	64 ± 5
k_ex_AC (s_^−1^_)_	418 ± 13	227 ± 14	169 ± 9
k_ex_BC (s_^−1^_)_	93 ± 15	~0	~0

**Table 2 viruses-14-00632-t002:** Populations and kinetic rate constants obtained from the fit of CPMG data or/and CEST data for NCp7 at 35 °C. The best model was a three-state model (in which A is the major form and B and C are the minor forms) with interconversion between the three states. Different sets of data have been used for these calculations: in the first column, CEST data are shown in combination with CPMG data for the residues H44, W37, C39. In the second column, CEST data are shown for the residues H44, W37, C39 with CPMG data for ten residues of ZK2; the third column comprises only the CPMG data (similarly to the calculations of Table 1). In the fourth column, only CEST data have been used. In the fifth column, CEST data for the residues H44, W37, C39, G40, and E42 with CPMG data for the residues H44, W37, C39 have been used.

	CEST_H44,W37,C39_ + CPMG_H44,W37,C39_	CEST_H44,W37,C39_ + CPMG_10residues_	CPMG_10residues_	CEST_H44,W37,C39_	CEST_H44,W37,C39,G40,E42_ + CPMG_H44,W37,C39_
p_A_	0.959 ± 0.001	0.957 ± 0.001	0.945 ± 0.002	0.967 ± 0.002	0.961 ± 0.001
p_B_	0.015 ± 0.001	0.016± 0.001	0.026 ± 0.002	0.013 ± 0.001	0.016 ± 0.001
p_C_	0.025 ± 0.001	0.026± 0.001	0.028 ± 0.001	0.019 ± 0.003	0.022 ± 0.001
k_ex_AB (s_^−1^_)_	462 ± 40	507 ± 42	349 ± 38	297 ± 72	504 ± 25
k_ex_AC (s_^−1^_)_	593 ± 45	524 ± 36	415 ± 24	149 ± 48	630 ± 24
k_ex_BC (s_^1^_))_	122 ± 56	10 ± 30	79 ± 24	347 ± 68	73 ± 32
Χ^2^reduced	1.77	1.51	1.12	1.20	2.43

## Data Availability

Not applicable.

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
