# Peer review of "Investigation of the Low-Populated Excited States of the HIV-1 Nucleocapsid Domain"

_viruses, 2022, doi:10.3390/v14030632_

Round 1

Reviewer 1 Report

Mouhand used advanced 15N NMR experiments to investigate the conformational dynamics of NCd. It is clear that not just NCp7, but NCp9 and NCp15 also show similar three state dynamics largely localized to the ZK2 region. The populations of the minor states are significant enough at 35 oC that they can be physiologically relevant. The paper is interesting but following questions should be addressed:

Major points:

  • Table 2: The reported exchange parameters are inconsistent with one another. For example the fits reported in column 3 (CPMG10residues) and column 4 (CEST H44,W37,C39) are quite good with reduced chi square of 1.1 and 1.2 respectively but the combined fit gives a higher reduced chi square 3.36, suggesting the datasets are incompatible with one another. Will removing bounds on R2B and R2C while fitting the CEST data make a difference?
  • It is clear that protein concentration has an effect on NCp7 dynamics. How are the rates and populations affected by varying protein concentration? Is one of the exchange process more sensitive to concentration? If so which one and why?

Minor points:

  • Line 124: How was R2(0) estimated?
  • Line 214: “To minimize ..” Increasing the pH will increase water exchange effects. The authors should clarify what they mean.
  • Figure 2 Legend, Lines 229,230: Fits to which model?
  • How do the DWAB and DWAC values compare with those obtained previously, for example in Ref 34.
  • Figure 4: Experimental data is not visible for G40, Q45 and W37.
  • Line 345: What are simulated R2 values?
  • Were any special experiments carried out to obtain the sign of DWAB and DWAC from CPMG data or were they obtained from the CEST data?

Author Response

Manuscript ID: viruses-1617511

Dear Editor,

We thank the two reviewers for their constructive comments. We are happy to submit a revised version of our manuscript and hope that it is now suitable for publication in Viruses in the special issue “Retroviral nucleocapsid protein”.

Below, we address each of the reviewers’ comments.

Sincerely,

Dr Carine Tisné

Dr Olivier Mauffret

Referee 1

Comments and Suggestions for Authors

Mouhand used advanced 15N NMR experiments to investigate the conformational dynamics of NCd. It is clear that not just NCp7, but NCp9 and NCp15 also show similar three state dynamics largely localized to the ZK2 region. The populations of the minor states are significant enough at 35 oC that they can be physiologically relevant. The paper is interesting but following questions should be addressed:

Major points:

  • Table 2: The reported exchange parameters are inconsistent with one another. For example the fits reported in column 3 (CPMG10residues) and column 4 (CEST H44, W37,C39) are quite good with reduced chi square of 1.1 and 1.2 respectively but the combined fit gives a higher reduced chi square 3.36, suggesting the datasets are incompatible with one another. Will removing bounds on R2B and R2C while fitting the CEST data make a difference?

We thank the referee for this remark. The value 3.36 pointed by the referee was a mistake. We apologize for this and we have carefully checked all the values in the Tables. The correct value is indeed 1.51 for the reduced chi-square. The value of 1.51 indicates a slight decrease in the fit of the data due to the fact that more data are included in the fit. However, this value is acceptable and shows that this fit leads to similar parameters for the description of the exchange processes.

  • It is clear that protein concentration has an effect on NCp7 dynamics. How are the rates and populations affected by varying protein concentration? Is one of the exchange process more sensitive to concentration? If so which one and why?

Yes, we clearly show that concentration has an effect on NCp7 dynamics, as seen in some 15N-CPMG profiles. The description of the exchange processes as a function of protein concentration would have required experiments recorded at other magnetic fields and a full analysis of the data with appropriate software such as ChemEx. Unfortunately, the low concentration of NCp7 and lower magnetic field results in experiments with poor signal-to-noise ratio that prevent the analysis of CPMG experiments with confidence.

Minor points:

  • Line 124: How was R2(0) estimated?

R2(0) is estimated using R2(vCPMG)=  , the value chosen for R2(0) is that for which the CPMG frequency is the lowest, i.e vCPMG =25, the first point we can measure.

  • Line 214: “To minimize ..” Increasing the pH will increase water exchange effects. The authors should clarify what they mean.

We agree that increasing the pH increases water exchange effects. We corrected the sentence to be clearer.

  • Figure 2 Legend, Lines 229,230: Fits to which model?

The fit was done with ChemEx for a model of three states, one major and two minor states that interconvert. This is now stated in the figure legend.

  • How do the DWAB and DWAC values compare with those obtained previously, for example in Ref 34.

The chemical shift differences DWAB and DWAC are extremely similar to that published by the group of Clore et al. and are confirmed by the CEST experiments. This allows us to discuss the models proposed for state B and C in this previous study for NCp9 and NCp15. A sentence has been added to make this point clearer.

  • Figure 4: Experimental data is not visible for G40, Q45 and W37.

Figure 4 has been corrected.

  • Line 345: What are simulated R2 values?

We called “simulated R2”, R2 that were evaluated from CEST experiments and fits using ChemEX. We have removed this term.

  • Were any special experiments carried out to obtain the sign of DWAB and DWAC from CPMG data or were they obtained from the CEST data?

The signs of DWAB and DWAC were obtained from the CEST data of NCp7 and the same signs were used to represent DWAB and DWAC of NCp9 and NCp15 in Figure 3. This is now made clear in the legend of Figure 3.

Reviewer 2 Report

The manuscript by Mauffret, Tisne, and coworkers describes the lowly populated transient states of the C-terminal zinc knuckle of HIV-1 nucleocapsid (NCp7). The major findings of this manuscript are as follows:

  1. The authors use chemical exchange-based NMR methods, 15N-CPMG experiments, to characterize the conformational exchange exhibited by the C-terminal zinc knuckle of NCp7. The following conditions were varied to understand their role in the said chemical exchange: temperature, ionic strength, and pH. The authors also carried out similar experiments on two larger fragments of nucleocapsid, NCp9 and NCp15. The NMR data were fit to a three-state model (A <=> B <=> C, where A is a major state and minor states B and C interconvert). They show that in all three constructs (NCp7, NCp9, and NCp15), the C-terminal zinc knuckle follows a millisecond exchange regime between these three states. In contrast, the N-terminal zinc knuckle likely exists in a single conformation.
  2. The authors carried out 15N-CEST experiments to confirm their findings, and a global fitting of CEST and CPMG data was performed to further characterize the chemical exchange exhibited by the C-terminal zinc knuckle.
  3. The authors measured 15N-CPMGs at two different concentrations of NCp7 to rule out the contribution of concentration-dependent oligomerization in the chemical exchange of the C-terminal zinc knuckle.

Overall, a large amount of NMR data is presented in this manuscript. There are, however, several issues associated with this study, as described below.

1) Lack of novelty: The minor states of NCp7 were uncovered and characterized by the Clore group (see Deshmukh et al., Angew. Chem. Int. Ed. 2018; the authors of the current study have accidentally cited this manuscript twice, reference no. 34 and 44). They showed that the C-terminal zinc knuckle exhibits two folded minor species in which one of the coordination bonds (Cys36-Zn or His44-Zn) is hydrolyzed, and that the antiretroviral thioesters exploit this exchange by acylation of Cys36. It is satisfying to see that the authors of the current study replicated many of the findings of the previous study. However, the conclusions of the current manuscript are not sufficiently novel as these are essentially the same findings as those of the Clore group.

2) NMR data and data fitting: It is puzzling to see why the authors report quantitative analyses of 15N-CPMG data (Fig. 1-3, Table 1), followed by global analyses of 15N-CEST and -CPMG data (Table 2). The chemical exchange of the C-terminal zinc knuckle is accessible to both CPMG and CEST NMR experiments. As such, the authors should only report the results of the global fit (CEST + CPMG) in the main text. It will also help to alleviate redundancy as the individual and global fits gave similar results (see Table 2).

The rate constants reported in Table 1 are not consistent between different nucleocapsid constructs. It is especially difficult to understand why states B and C do not interconvert in the case of NCp9 and NCp15. The changes in the population of each species in these two constructs compared to NCp7 may therefore arise from a discrepancy in data fitting and not because of additional transient interactions as claimed in this study.

The authors report reduced chi2 in Table 2. However, it is difficult to judge their significance as different datasets are used for fitting.

The authors should discuss how they determined the sign of the chemical shift differences between the major and minor states using only CPMGs (Fig. 3 and S2).

It is unclear as to what is represented in Figure 4. Are these experimental 15N-CEST data and the corresponding fits (as mentioned in the caption)? This is because no actual data points are shown and only the Cys39 panel shows error bars. Incidentally, what are these error bars? Are these uncertainties in data or do these errors arise from data fits? In either case, it is difficult to understand why errors are shown only for 1 residue (and not for any of the other reported datasets, namely Fig. 2, 3, 4, and S2). The authors should also discuss how these CEST data were normalized (i.e., were these normalized against a reference dataset acquired using a TCEST period but a B1 field strength of 0 Hz?).

I am also puzzled by the bar graphs shown in Fig. 2B. How do the authors assign the backbone resonances of residues of the flexible N-termini of NCp9 and NCp15? For instance, in the case of NCp9, the authors show only a single residue (most likely residue 2; it is not possible to accurately determine which residue has been reported here based on the data presented in this figure). For NCp15, the authors report two additional residues (9 and 10). It is difficult to accurately assign the 1H-15N cross-peak of a single isolated residue using backbone NMR experiments. Were these residues assigned using the other datasets collected at other experimental conditions? If that is true, then how reliable are these assignments?

There is also some discrepancy in the CPMG curves of residue His44 at 950 MHz (Figure 2C). Can the authors comment on why this particular residue does not follow the same trend at 950 MHz as opposed to 700 MHz?

How is it possible that the authors get bigger dispersions for the following residues, Trp37, Cys39, and Gly40, at 700 MHz (Fig. S2) than at 950 MHz (Fig. 5)?

3) Presentation: The manuscript is quite difficult to follow. For instance, the kinetic scheme used for data fitting is not mentioned in the main text but is placed in the SI (Fig. S1). It should be in the main text to guide the reader. The manuscript can also be shortened considerably. Moreover, the authors should proofread the manuscript thoroughly for grammatical and punctuation errors.

Author Response

Manuscript ID: viruses-1617511

Dear Editor,

We thank the two reviewers for their constructive comments. We are happy to submit a revised version of our manuscript and hope that it is now suitable for publication in Viruses in the special issue “Retroviral nucleocapsid protein”.

Below, we address each of the reviewers’ comments.

Sincerely,

Dr Carine Tisné

Dr Olivier Mauffret

Referee 2

Comments and Suggestions for Authors

The manuscript by Mauffret, Tisne, and coworkers describes the lowly populated transient states of the C-terminal zinc knuckle of HIV-1 nucleocapsid (NCp7). The major findings of this manuscript are as follows:

  1. The authors use chemical exchange-based NMR methods, 15N-CPMG experiments, to characterize the conformational exchange exhibited by the C-terminal zinc knuckle of NCp7. The following conditions were varied to understand their role in the said chemical exchange: temperature, ionic strength, and pH. The authors also carried out similar experiments on two larger fragments of nucleocapsid, NCp9 and NCp15. The NMR data were fit to a three-state model (A <=> B <=> C, where A is a major state and minor states B and C interconvert). They show that in all three constructs (NCp7, NCp9, and NCp15), the C-terminal zinc knuckle follows a millisecond exchange regime between these three states. In contrast, the N-terminal zinc knuckle likely exists in a single conformation.
  2. The authors carried out 15N-CEST experiments to confirm their findings, and a global fitting of CEST and CPMG data was performed to further characterize the chemical exchange exhibited by the C-terminal zinc knuckle.
  3. The authors measured 15N-CPMGs at two different concentrations of NCp7 to rule out the contribution of concentration-dependent oligomerization in the chemical exchange of the C-terminal zinc knuckle.

Overall, a large amount of NMR data is presented in this manuscript. There are, however, several issues associated with this study, as described below.

1) Lack of novelty: The minor states of NCp7 were uncovered and characterized by the Clore group (see Deshmukh et al., Angew. Chem. Int. Ed. 2018; the authors of the current study have accidentally cited this manuscript twice, reference no. 34 and 44). They showed that the C-terminal zinc knuckle exhibits two folded minor species in which one of the coordination bonds (Cys36-Zn or His44-Zn) is hydrolyzed, and that the antiretroviral thioesters exploit this exchange by acylation of Cys36. It is satisfying to see that the authors of the current study replicated many of the findings of the previous study. However, the conclusions of the current manuscript are not sufficiently novel as these are essentially the same findings as those of the Clore group.

We agree that we found the same minor states of NCp7 than those described in the previous work of Deshmukh et al. 2018. However, our study added a considerable amount of data on the three maturation forms of the nucleocapsid domain (NCd) instead of only the mature one (NCp7). We thus think that studies of both groups are complementary. We also believe that studying all maturation forms of NCd in a comparative study is essential. We directly show that NCp15 and NCp9 populate minor states and this validates the model of targeting minor states of NCd. Indeed, several publications (Jenkins et al., 2010; Miller Jenkins et al., 2019) show that the Gag polyprotein, where NCd is in a context similar to NCp9 and NCp15, is targeted by mercaptobenzamide thioester perturbing the protein maturation. If the targeting mechanism of these inhibitors is really the one proposed by Desmukh et al., with specific targeting of the minor species, it is thus necessary that Gag, NCp15 and NCp9 also populate these minor states, what we show here for the first time. Our study also reveals the essential role of temperature in the appearance of minor states, a feature not previously anticipated or studied.

Both studies use recent NMR methodologies for which quantitative evaluations and robustness still need validations. We think that this study confirms the robustness of these methods and is far helpful for that.

2) NMR data and data fitting: It is puzzling to see why the authors report quantitative analyses of 15N-CPMG data (Fig. 1-3, Table 1), followed by global analyses of 15N-CEST and -CPMG data (Table 2). The chemical exchange of the C-terminal zinc knuckle is accessible to both CPMG and CEST NMR experiments. As such, the authors should only report the results of the global fit (CEST + CPMG) in the main text. It will also help to alleviate redundancy as the individual and global fits gave similar results (see Table 2).

The referee missed that Figs.1-3 and Table 1 reports quantitative analyses for NCp7, NCp9 and NCp15 the three maturation forms of NCd, whereas Table 2 reports analyses for NCp7 only. The rationale for that is that 1) first, we show that the three maturation forms of NCd populate minor states and that these minor states are similar whatever the maturation form, 2) we used CEST + CPMG data on NCp7 to bring details on the dynamics of these forms. We thought that it was not necessary to measure all these NMR experiments for NCp7, NCp9 and NCp15 given that the minor states populated by these proteins are the same whatever the maturation step.

The rate constants reported in Table 1 are not consistent between different nucleocapsid constructs. It is especially difficult to understand why states B and C do not interconvert in the case of NCp9 and NCp15. The changes in the population of each species in these two constructs compared to NCp7 may therefore arise from a discrepancy in data fitting and not because of additional transient interactions as claimed in this study.

We made additional calculations to evaluate the influence of the magnitude of the interconversion rates between minor states on the other fitted parameters like the populations (see the table above).

CPMG10residues

CESTH44,W37,C39

CPMG10residues,

kex_BC=0

CPMG10residues,

kex_AC=0

CESTH44,W37,C39

kex_BC=0

pA

0.945 ± 0.002

0.967 ± 0.002

0.950 ± 0.002

0.27 ± 0.149

0.59

pB

0.026 ± 0.002

0.013 ± 0.001

0.023 ± 0.002

0.006 ± 0.004

0.008

pC

0.028 ± 0.001

0.019 ± 0.003

0.027 ± 0.001

0.71± 0.153

0.38

kex_AB (s-1)

349 ± 38

297 ± 72

421 ± 36

659 ± 60

485 ± 31

kex_AC (s-1)

415 ± 24

149 ± 48

447 ± 23

0

115 ± 4

kex_BC (s1))

79 ± 24

347 ± 68

0

650 ± 64

0

Χ22reduced

1.17

1.20

1.15

2.77

1.30

Χ22Total

441

792

434

1045

854

The calculations have been carried out in the same conditions used for Table 2. The two first columns of the table consider the calculations with only CPMG or only CEST data. In the third column, kex_BC is fixed to 0 and no degradation of the fit is observed (while one variable has been removed). The fit is even better (chi2 of 434 vs 441) and the parameters are only slightly modified. In contrast, the same calculations conducted with kex_AC=0 (fourth column) or with CEST data and kex_BC=0 (fifth column) led to unrealistic values of the parameters, notably pC. In our opinion, these simulations show that CPMG data are poorly sensitive to kex_BC rate constant and can be fitted without loss in quality of fit (F-test is even unnecessary since chi2 decrease with the suppression of kex_BC) and with only slight modifications of parameters: kex_AB shows a slight increase. In contrast, dramatic alterations of the parameters are observed when the CEST data are considered showing their sensitivity to the rate of interconversion between the minor states.

In conclusion, we believe that the kex values in Table 1 cannot really be estimated from the CPMG data and that this feature does not interfere with the population assessment. We have been cautious in discussing this table and only discuss the relative population of the different states for the three proteins. The text has been clarified on this point.

The authors report reduced chi2 in Table 2. However, it is difficult to judge their significance as different datasets are used for fitting.

The Chi2 reported in Table 2 are not the true Chi2 (

 (see materials and methods for details) but the Chi2_reduced that is the previous value divided by the difference between the number of data points and the number of variables (i.e. the number of degrees of freedom). We think that the Chi2_reduced can be used to compare the agreement between modelled and experimental data when comparing different data sets with a difference in the number of data points and the number of variables.

The authors should discuss how they determined the sign of the chemical shift differences between the major and minor states using only CPMGs (Fig. 3 and S2).

We agree that the signs of chemical shift differences between the major and minor states are difficult to extract using only CPMG data (Skynnikov, Dahlquist, Kay, 2002). These signs are however immediately apparent on the CEST data we recorded for NCp7. For NCp9 and NCp15, we used the same signs determined for NCp7 with the CEST data. This is now added in the legend of Figure 3.

It is unclear as to what is represented in Figure 4. Are these experimental 15N-CEST data and the corresponding fits (as mentioned in the caption)? This is because no actual data points are shown and only the Cys39 panel shows error bars. Incidentally, what are these error bars? Are these uncertainties in data or do these errors arise from data fits? In either case, it is difficult to understand why errors are shown only for 1 residue (and not for any of the other reported datasets, namely Fig. 2, 3, 4, and S2). The authors should also discuss how these CEST data were normalized (i.e., were these normalized against a reference dataset acquired using a TCEST period but a B1 field strength of 0 Hz?).

Figure 4 displays CEST data of some NCp7 residues. The data points and error bars are shown together with the fit made with ChemEx. The lack of error bars for most of the residues is an oversight that we have corrected in the new figure. The CEST errors stem from the error peak height due to random noise, estimated by parabolic interpolation with nmrPipe. In a conservative approach, the error range was fixed to three times this value. The CEST data were normalized with a data set acquired using a CEST period but with B1 field strength set to 0.

I am also puzzled by the bar graphs shown in Fig. 2B. How do the authors assign the backbone resonances of residues of the flexible N-termini of NCp9 and NCp15? For instance, in the case of NCp9, the authors show only a single residue (most likely residue 2; it is not possible to accurately determine which residue has been reported here based on the data presented in this figure). For NCp15, the authors report two additional residues (9 and 10). It is difficult to accurately assign the 1H-15N cross-peak of a single isolated residue using backbone NMR experiments. Were these residues assigned using the other datasets collected at other experimental conditions? If that is true, then how reliable are these assignments?

The assignment of the NMR backbone resonances of NCp7, NCp9 and NCp15 were performed at 10°C (BMRB entry number 26843, Larue,V. et al. (2018) Biomol. NMR Assign.), using classical 3D NMR experiments (HNCA, HN(CO)CA, CBCA(CO)NH, HNCACB, HNCO, HN(CA)CO, NOESY and TOCSY). Concerning the assignment of the amide group at 35°C, they were carefully followed by increasing the temperature by 5°C from 10°C to 35°C in TROSY experiment. Residues that were difficult to monitor to 35°C and/or for which they were superimposed with other residues at this temperature were not used for the analysis and so were not shown in the figure 2B. This is why we have data for residue 2 or not for the neighbouring residues.

There is also some discrepancy in the CPMG curves of residue His44 at 950 MHz (Figure 2C). Can the authors comment on why this particular residue does not follow the same trend at 950 MHz as opposed to 700 MHz?

We checked these data and we did not identify any error. H44 is the residue that exhibits the higher dispersion. It is possible that, at 950 MHz, the Rex becomes extremely high leading to high difference of frequency between major and minor states at 950 MHz, this situation could lead to difficulties to suppress chemical exchange contributions at CPMG frequencies of 1000 Hz, possibly higher frequencies could be necessary.

How is it possible that the authors get bigger dispersions for the following residues, Trp37, Cys39, and Gly40, at 700 MHz (Fig. S2) than at 950 MHz (Fig. 5)?

The referee has indeed observed effectively an unusual behavior of the CPMG data since it is not expected that bigger dispersion (larger Rex) could be obtained at lower frequency (700 vs 950 MHz). However, it is quite clear that it is what we observe for C39 and possibly for G40 but not for W37 (dispersion is still higher at 950 than at 700 MHz). Indeed, in the present case, the data of the two figures from which the referee is talking about have been recorded in significantly different conditions: the samples are different, with concentrations of 1mM and 400 μM, as described in materials and methods the level of D2O is different in the experiments of Fig5 vs those of Figure S2 (10% and 4% respectively) and also the number of CPMG points are different. If we now consider the 700 and 950 MHz data recorded under exactly the same conditions, we never observe a higher dispersion for any residue at 700 than at 950 MHz.

3) Presentation: The manuscript is quite difficult to follow. For instance, the kinetic scheme used for data fitting is not mentioned in the main text but is placed in the SI (Fig. S1). It should be in the main text to guide the reader. The manuscript can also be shortened considerably. Moreover, the authors should proofread the manuscript thoroughly for grammatical and punctuation errors.

The manuscript has been edited and we have decided not to shorten it as we show the results for three proteins and compare the different fitting procedures, which is the core of this paper.

Round 2

Reviewer 1 Report

Please accept

Reviewer 2 Report

See prior review.  Authors have addressed my questions.